# CODA: COMMONSENSE-DRIVEN AUTOREGRESSIVE HUMAN INTERACTION GENERATION

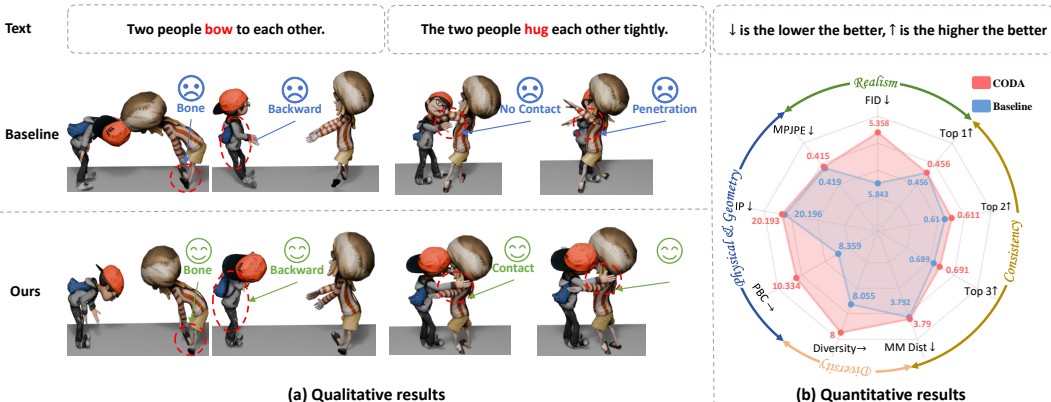

Figure 1: Performance shows of CODA. (a) Qualitative results compared with the baseline (*i.e.*, InterMask (Javed et al., 2025)). Our method demonstrates strong commonsense plausibility (*e.g.*, fixed bone lengths, correct body structures, realistic contact, and no penetration). (b) Quantitative results. Our CODA achieves better performance. "→" means the closer to real motion the better.

## ABSTRACT

Human interaction generation (HIG) aims to synthesize commonsense-plausible interaction motion from textual descriptions. However, most existing generation methods as diffusion and autoregressive models typically overlook explicit commonsense constraints, leading to implausible motion artifacts such as bone stretching or penetration. To address these issues, this work proposes a novel learning paradigm **CODA** with two core components: Interactive Codebook Storager (ICS) and Commonsense Constraint Loss (CCL). Specifically, ICS captures and stores commonsense features of single-person motion and human-human interaction, ensuring high-quality motion generation. Based on this, CCL constrains single-person joint trajectories, regulates the center-of-mass position, and applies distance and collision constraints in multi-person interactions, effectively suppressing motion artifacts and explicitly enforcing commonsense plausibility. Extensive experimental results suggest that our CODA generates higher-quality HIG scenarios than existing state-of-the-art methods.

## 1 INTRODUCTION

Human Interaction Generation (HIG) strives to synthesize motion sequences that are not only commonsense plausible but also naturally fluent (Yang et al., 2024; Li et al., 2025b; Guo et al., 2022). This capability makes it profoundly valuable for many applications such as games, robotics, and VR/AR (Li et al., 2025a; Qu et al., 2024; 2025; Xing et al., 2024). This practical demand has stimulated considerable studies on the generation of human motion sequences with textual descriptions (Guo et al., 2024), especially for single-person scenarios (Chen et al., 2023; Zhang et al., 2025; Li et al., 2024; Bian et al., 2025; Tevet et al., 2022). However, many practical application scenarios require describing and demonstrating the process of multi-person interaction. To satisfy it, mainstream approaches of HIG are divided into two categories: conditional diffusion models (Cai et al., 2024;

Liang et al., 2024; Ruiz-Ponce et al., 2024; Wang et al., 2024; Fan et al., 2025) and mask-based autoregressive models (Javed et al., 2025). These models have demonstrated promising capabilities in capturing the intricate multimodal distribution of motions, thereby significantly enhancing the diversity and quality of motion generation.

Despite achieving advancements, most existing approaches (Javed et al., 2025; Liang et al., 2024) fail to address a crucial aspect of human motion synthesis: commonsense plausibility, *i.e.*, consistency with both physical laws and common human interaction patterns. Concretely, these techniques excel in capturing the correlations between textual semantics and motion features, but they predominantly rely on implicit representations of commonsense constraints. As a result, executing the *"two people bow to each other"* gives rise to individual counterintuitive motions (*e.g.*, elongated legs and backward leaning), as shown in Fig. 1(a). Moreover, executing the *"the two people hug each other tightly"* leads to interactive counterintuitive motions (*e.g.*, missing contact and penetration). Considering humans' remarkable sensitivity to even the most minute deviations, motions that lack commonsense plausibility pose significant obstacles to their deployment in real-world applications (Li et al., 2025b).

For improving commonsense plausibility, PhysDiff (Yuan et al., 2023) and Morph (Li et al., 2025b) employ reinforcement learning with physics simulators (*e.g.*, IsaacGym) to enforce physical commonsense in single-person motion generation. Moreover, Stablemodiffusion (Huang et al., 2024) and MoMask (Guo et al., 2024) incorporate geometric commonsense through foot-sliding and pose losses to prevent unrealistic foot movements in single-person motion generation. Despite the progress made, these methods encounter several key limitations: 1) the frequent simulator invocations increase computational overhead; 2) a lack of plug-and-play commonsense plausible constraints specifically designed for interactive motion generation tasks leads to motion artifacts such as bone stretching and penetration. A natural question arises: ***Is it possible to design a plug-and-play commonsense plausible model without relying on simulators?***

To answer the above question, we present a novel framework **CODA** with commonsense-guided vector quantized variational autoencoder, addressing commonsense implausible motion artifacts in interactive motion generation. CODA mainly includes two core components: the Interactive Codebook Storager (ICS) and the Commonsense Constraint Loss (CCL). Specifically, ICS captures and stores commonsense features from both single-person and two-person interactions at the feature level to avoid the frequent simulator invocations on post-processing. During the motion generation stage, commonsense plausible motions can be efficiently generated by simply performing a fast lookup in the codebook. Furthermore, we designed the plug-and-play CCL to guide ICS in learning commonsense-consistent features. CCL is composed of both single-person (*e.g.*, center of mass loss and key joint trajectory loss) and human-human losses (*e.g.*, Gaussian joint distance map loss and penetration penalty loss). Specifically, the center of mass loss mitigates backward-leaning issues by regulating body posture through the computation of the center of mass and its angle relative to the ground. The key joint trajectory loss addresses unnatural bone stretching by ensuring accurate trajectories of critical joints (*e.g.*, hands, feet, and pelvis). Meanwhile, the Gaussian joint distance map loss and the penetration penalty loss tackle non-contact and interpenetration problems, respectively, using a soft-threshold distance strategy and a collision detection method. Building on this, we train a Conditional Masked Transformer (CMT) to align text descriptions with motion sequences using a masked prediction strategy. During inference, we adopt an autoregressive strategy to iteratively predict the masked tokens and generate commonsense-plausible motions. Extensive experiments demonstrated the effectiveness of our CODA and outperforms the state-of-the-art methods, including realism, consistency, diversity, physical plausibility, and geometric plausibility (see Fig. 1(b)).

Our main contributions are threefold:

- **New Interaction Motion Generation Framework.** We propose an autoregressive interactive motion generation framework (**CODA**) to capture individual and human-human commonsense features from the feature storage and commonsense loss constraints.

- **Interactive Feature Storage.** We propose an interactive codebook storage that captures single-person motion and two-person interaction commonsense features, effectively addressing the issue of limited interaction capability in a single-codebook method.

- **Commonsense Constraint Loss.** We propose a commonsense constraint loss with hierarchical constraints for single-person and human-human scenarios, effectively addressing issues such as bone stretching, backward leaning, non-contact, and penetration.

Figure 2: Overview of CODA. CODA is a two-stage autoregressive framework for HIG generation. In Stage I, commonsense-guided vector quantized variational autoencoder encodes individual and interactive motion using Interactive Codebook Storager (ICS) under Commonsense Constraint Loss (CCL) supervision (§3.1). CCL enforces commonsense plausibility via four components: $\mathcal{L}_{\text{KTraj}}$ prevents bone stretching, $\mathcal{L}_{\text{COM}}$ restricts backward leaning, $\mathcal{L}_{\text{GDM}}$ addresses non-contact via soft-thresholding, $\mathcal{L}_{\text{PPL}}$ penalizes body penetration (§3.1). In Stage II, a Conditional Masked Transformer (CMT) aligns textual semantics with motion sequences via the Inter-M Transformer (§3.2).

## 2 RELATED WORK

**Human-Human Interaction Generation.** Human-Human Interaction generation aims to produce natural and realistic motion sequences from textual descriptions. Most research on human interaction generation has focused on two-person interactions, with two primary objectives: i) reaction generation (Cai et al., 2024; Xu et al., 2024; Ruiz-Ponce et al., 2024), which involves generating the reactor's motion in response to the actor's motion. For instance, in2IN (Ruiz-Ponce et al., 2024) improves the diversity of interactive motion generation by integrating role-specific action descriptions with individual priors. ii) interaction generation (Javed et al., 2025; Shafir et al., 2023; Fan et al., 2025), which involves simultaneously generating the motions of both individuals to ensure a coherent and dynamic interaction. For example, InterMask (Javed et al., 2025) leverage human identity symmetry to generate high-quality interaction motions through autoregressive models. However, these methods struggle to generate commonsense-plausible motions, as they overlook explicit commonsense constraints during training. In contrast, our proposed CODA effectively incorporates commonsense constraints, enhancing the commonsense plausibility of the generated motions.

**Commonsense plausibility.** The goal of commonsense plausibility is to ensure that generated motions conform not only to physical laws but also to natural human interaction patterns. Ensuring commonsense plausibility in motion generation remains a challenging task. To enhance the plausibility, recent studies can be divided into two major groups: i) Physical commonsense plausibility (Li et al., 2025b; Yuan et al., 2023; Han et al., 2024), which improves the realism of generated motions via a physics-based reinforcement learning approach; and ii) Geometric commonsense plausibility (Huang et al., 2024; Guo et al., 2024), which integrates a geometric prior via foot-sliding and pose constraint losses to strengthen motion plausibility. Despite their success, frequent use of the simulator increases computational overhead and lacks *plug-and-play* commonsense plausibility constraints specifically designed for interactive motion generation. This leads to implausible motion generation, such as bone stretching and penetration. In contrast, our CCL enhances the plausibility of interactive motions through hierarchical constraints without relying on a simulator, and it is plug-and-play.

## 3 METHODS

Our goal is to generate the interaction motion of two people given a textual description, denoted as $\{\boldsymbol{m}_p\}_{p \in \{a,b\}}$. Here, $\boldsymbol{m}_p \in \mathbb{R}^{N \times J \times d}$ represents the motion sequence of an individual (either $a$ or $b$), consisting of $N$ poses, each with $J$ joints and $d$-dimensional joint features. As illustrated in Fig. 2, our CODA consists of two stages. In the first stage, Commonsense-guided Vector Quantized Variational AutoEncoder (CVQ-VAE) extracts and stores individual and interaction commonsense features through ICS under the guidance of CCL (§3.1). In the second stage, building on CVQ-VAE, the CMT leverages a masking strategy and feature alignment to ensure text-motion consistency (§3.2). The inference process of generation is detailed (§3.3).

## 3.1 COMMONSENSE GUIDED VECTOR QUANTIZED-VARIATIONAL AUTOENCODER

In interactive motion generation (Javed et al., 2025; Liang et al., 2024), diffusion-based methods generate motion through iterative denoising but often produce unnatural and commonsense implausible results. In contrast, the autoregressive method utilizes a single motion codebook to store motion information, thereby improving the generation quality. However, as shown in Fig. 1(a), we observe that relying solely on a single codebook limits the ability to capture implausible interactive motion, leading to artifacts such as bone scaling, backward leaning, lack of contact, and body penetration.

**CVQ-VAE.** To address these challenges, we propose a CVQ-VAE, including two core components: an Interactive Codebook Storager (**ICS**) and a Commonsense Constraint Loss (**CCL**). Specifically, ICS effectively captures and stores both individual motion and interaction features, while CCL introduces individual and interaction constraints to guide the learning of commonsense information, improving the commonsense plausibility of motions.

**ICS.** Inspired by the shared codebook storage strategy for high-level semantics and low-level details proposed in UniTok (Ma et al., 2025), we design ICS to extract and store single-person and interaction features separately using pose ($\mathcal{C}$) and interaction ($\mathcal{P}$) codebooks. As illustrated in Fig. 2, given an input motion sequence $\boldsymbol{x}_p$, we first project it into the latent space $\tilde{\boldsymbol{t}}_{p \in \{a,b\}} \in \mathbb{R}^{n \times j \times d'}$ using a shared ResNet-based encoder ($\varphi$). Each $d'$-dimensional latent feature is then quantized by replacing it with its nearest neighbor in the single-person pose codebook $\mathcal{C} = \{\boldsymbol{c}_k\}_{k=0}^{|\mathcal{C}|-1}$, resulting in the quantized sequence $\boldsymbol{t}_p = \mathcal{Q}(\tilde{\boldsymbol{t}}_p)$. $\mathcal{Q}$ denotes the vector quantization. Then, we utilize self-attention and cross-attention to extract and capture interactive motion features:

$$\tilde{\boldsymbol{z}}_a = \psi_a(\boldsymbol{t}_a), \quad \tilde{\boldsymbol{z}}_b = \psi_b(\boldsymbol{t}_b), \quad \check{\boldsymbol{z}}_a = \phi_a(\tilde{\boldsymbol{z}}_a, \tilde{\boldsymbol{z}}_b), \quad \check{\boldsymbol{z}}_b = \phi_b(\tilde{\boldsymbol{z}}_b, \tilde{\boldsymbol{z}}_a), \tag{1}$$

where $\psi$ denotes the self-attention, and $\phi$ represents the cross-attention. Subsequently, we replace $\check{\boldsymbol{z}}_p$ with the closest entry in the interaction codebook $\mathcal{P} = \{\rho_k\}_{k=0}^{|\mathcal{P}|-1}$, generating the quantized sequence $\boldsymbol{z}_p = \mathcal{Q}(\check{\boldsymbol{z}}_p)$. Next, we integrate individual and interactive information through vector summation:

$$\boldsymbol{w}_a = \boldsymbol{z}_a + \boldsymbol{t}_a, \quad \boldsymbol{w}_b = \boldsymbol{z}_b + \boldsymbol{t}_b, \tag{2}$$

finally, the pose decoder ($\hat{\varphi}$) and commonsense decoder ($\tilde{\varphi}$) reconstruct the pose motion $\hat{\boldsymbol{m}}_p = \hat{\varphi}(\boldsymbol{w}_p)$ and commonsense motion $\hat{\boldsymbol{z}}_p = \tilde{\varphi}(\boldsymbol{z}_p)$, respectively. At this stage, the training objective of our CVQ-VAE consists of two motion reconstruction losses and two commitment losses:

$$\mathcal{L}_{\text{cvq}} = \|\boldsymbol{m}_p - \hat{\boldsymbol{m}}_p\|_1 + \alpha \|\tilde{\boldsymbol{t}}_p - \text{sg}(\boldsymbol{t}_p)\|_2^2 + \|\boldsymbol{z}_p - \hat{\boldsymbol{z}}_p\|_1 + \beta \|\check{\boldsymbol{z}}_p - \text{sg}(\boldsymbol{z}_p)\|_2^2, \tag{3}$$

where $\text{sg}(\cdot)$ denotes the stop-gradient operation, and $\alpha$ and $\beta$ is a weighting factor. We use Exponential Moving Average (EMA) and codebook reset to update $\mathcal{C}$ and $\mathcal{P}$.

**CCL**. To enhance the commonsense plausibility, we propose the CCL, which explicitly guides ICS in extracting and storing commonsense-plausible features. CCL consists of two components: 1) *Center of mass loss* and *Key joint trajectory loss* correct backward-leaning and unnatural bone stretching by regulating body posture and critical joint trajectories, respectively; 2) *Gaussian joint distance map loss* and *Penetration penalty loss* respectively address non-contact and penetration issues through a soft-threshold distance and collision detection method.

*Center of mass loss.* From a biomechanical perspective, the human body can self-adjust during movement, ensuring that the center of mass (COM) remains within a stable region to prevent losing balance. If the COM shifts too far backward, the body must compensate through the feet or torso, which can lead to unnatural body postures (Zhang et al., 2022). Therefore, we propose a COM loss function based on distance and angle, which penalizes poses that violate biomechanical principles, ensuring that the generated motions are more commonsense plausible. The formula is as follows:

$$\mathcal{L}_{\text{COM}} = \|\text{C}(\hat{\boldsymbol{m}}_p) - \text{C}(\boldsymbol{m}_p)\|_2^2 + \|\text{A}(\hat{\boldsymbol{m}}_p) - \text{A}(\boldsymbol{m}_p)\|_2^2, \tag{4}$$

where $\text{C}$ denotes the distance from COM to pelvis joint, and $\text{A}$ denotes the cosine of the angle between the line connecting the pelvis joint and the COM and the horizontal plane, reflecting the angle variation. Moreover, since studies (Zell et al., 2017) of human joints primarily focus on the relative motion and interactions between joints rather than absolute forces, the mass of each joint is set as 1. At this point, $COM = \frac{1}{J} \sum_{j=1}^{J} JP$. $JP$ denotes the coordinates of the joint $j$ in 3D space.

*Key joint trajectory loss.* During the generation of interactive motions from textual instructions, physical contact between the hands and other individuals, as well as between the feet and the ground,

introduces forces and reaction forces that influence skeletal positions (Zhang et al., 2022; Słowiński et al., 2016). These effects lead to incorrect codebook index selection for contact joints, resulting in unnatural bone stretching or compression. To alleviate this issue, we design a key joint trajectory loss, which measures the distance between the predicted and ground-truth trajectories of key joints on the XZ plane, thereby ensuring fixed bone lengths and commonsense plausibility of end-effectors in the generated motions. The formula is as follows:

$$\mathcal{L}_{\text{KTraj}} = \frac{1}{T} \sum_t ||\hat{\boldsymbol{m}}_k(t) - \boldsymbol{m}_k(t)||_1, \tag{5}$$

where t is the frame index at time t. $\hat{\boldsymbol{m}}_k$ and $\boldsymbol{m}_k$ denotes the motion features of $\hat{\boldsymbol{m}}_p$ and $\boldsymbol{m}_p$ after extracting the key joints $k$ (*i.e.*, hands, feet, and pelvis joints) as described in (Wan et al., 2024).

By incorporating $\mathcal{L}_{\text{COM}}$ and $\mathcal{L}_{\text{KTraj}}$, our model effectively mitigates issues such as backward leaning and bone stretching, significantly enhancing the quality of single-person motion generation. However, in interactive scenarios, noncontact cues such as speech and facial expressions are insufficient; commonsense interactions like hugging, handshaking, and pushing/pulling are also essential. As shown in Fig. 1(a), when executing the *"hug"* instruction, InterMask exhibits commonsense-implausible motions, *e.g.*, non-contact and severe interpenetration.

*Gaussian joint distance map loss.* To address the non-contact issue, we propose a Gaussian joint distance map (GDM) loss at the joint level, aimed at guiding the model to pay greater attention to the local interactive relationships between individuals. Specifically, we first compute the Euclidean distances between each pair of joints across individuals, denoted as the predicted distances ($\boldsymbol{D}_{ij}^{\text{pred}}$) and ground-truth distances ($\boldsymbol{D}_{ij}^{\text{gt}}$). Then, we employ a Gaussian weighting mechanism to compute soft-threshold weights for the distances between predictions and ground truth, enabling bidirectional supervision of whether contact occurs between individuals. The formula is as follows:

$$\mathcal{L}_{\text{GDM}} = \frac{1}{\sum_{i,j} \boldsymbol{w}_{ij}^{\text{pred}} + \epsilon} \sum_{i,j} \boldsymbol{w}_{ij}^{\text{pred}} ||\boldsymbol{D}_{ij}^{\text{pred}} - \boldsymbol{D}_{ij}^{\text{gt}}||_1 + \frac{1}{\sum_{i,j} \boldsymbol{w}_{ij}^{\text{gt}} + \epsilon} \sum_{i,j} \boldsymbol{w}_{ij}^{\text{gt}} \boldsymbol{D}_{ij}^{\text{pred}}, \tag{6}$$

where the first term guides local optimization with predicted weights, emphasizing the model focus on restoring distances in the areas it deems important. The second term reinforces the commonsense consistency of the perceived region with ground-truth weights, preventing the occurrence of uncontacted issues. $\epsilon = 1e^{-7}$ is a small constant added to the denominators to avoid division by zero. $\boldsymbol{w}_{ij}^* = \exp\left(-(\boldsymbol{D}_{ij}^*)^2 / 2\tau^2\right)$ denotes the Gaussian weighting mechanism. It assigns higher weights to joint pairs that are closer in distance, while the weights of distant pairs quickly decay, thereby focusing on the potential contact areas. Here, $\tau = 0.5$ is a temperature coefficient. The loss converges to 0 when predicted distances match the ground truth and all required contacts are perfectly achieved.

*Penetration penalty loss.* Since most existing interactive motion generation methods (Javed et al., 2025; Liang et al., 2024) lack explicit commnonsense constraints, the generated motions often produce penetration artifacts. To address this, we propose a body-level Penetration Penalty Loss (PPL). First, we detect spatial overlaps between body parts (*e.g.*, left/right legs, left/right arms, and spine) using Axis-Aligned Bounding Boxes (AABB). If an overlap is detected, we then compute the minimum bone-to-bone distance between each pair of body-part chains. When the predicted distance $\boldsymbol{d}^{\text{pred}}$ is below the ground truth minimum distance $\boldsymbol{d}^{\text{gt}}$, the loss penalizes the predicted values for being too small, while also introducing a mean squared error term for the minimum distance to maintain a reasonable distance distribution. The formula is as follows:

$$\mathcal{L}_{\text{PPL}} = \frac{1}{\text{CH}} \sum_{\text{ch}=1}^{\text{CH}} \mathbb{1}_{\text{overlap}}^{\text{ch}} \times \left[ \max\left(0,\, 0.9 \times \boldsymbol{d}_{\text{ch}}^{\text{gt}} - \boldsymbol{d}_{\text{ch}}^{\text{pred}}\right) + \left(\boldsymbol{d}_{\text{ch}}^{\text{pred}} - \boldsymbol{d}_{\text{ch}}^{\text{gt}}\right)^2 \right], \tag{7}$$

where $\mathbb{1}_{\text{overlap}}$ indicates whether an AABB overlap occurs, and CH is the number of valid bone chain pairs. The loss is averaged across all considered body part pairs to ensure detailed modeling and effective constraint of potential inter-body penetration motion.

**Overall Training Objective.** We jointly optimize the CVQ-VAE training loss and CCL (*cf.*, Algorithmic 1 ) by weighting parameters, enhancing generation motions commonsense plausibility:

$$\mathcal{L}_{\text{CCL}} = \lambda_{\text{COM}} \mathcal{L}_{\text{COM}} + \lambda_{\text{KTraj}} \mathcal{L}_{\text{KTraj}} + \lambda_{\text{GDM}} \mathcal{L}_{\text{GDM}} + \lambda_{\text{PPL}} \mathcal{L}_{\text{PPL}}, \tag{8}$$

$$\mathcal{L}_{\text{reg}} = \mathcal{L}_{\text{cvq}} + \mathcal{L}_{\text{CCL}}, \tag{9}$$

where $\lambda_{\text{COM}}$, $\lambda_{\text{KTraj}}$, $\lambda_{\text{GDM}}$, and $\lambda_{\text{PPL}}$ are the weight balance parameters.

### 3.2 CONDITIONAL MASKED TRANSFORMER

As shown in Fig. 2, we employ a Conditional Masked Transformer (CMT) with a masking strategy to capture the motion tokens of two individuals. Specifically, the features extracted by the encoder are discretized using a pose codebook to obtain token representations $\{t_a, t_b\}$. These token representations are then fused through stretching and concatenation. A randomly applied masking strategy, controlled by a cosine scheduling function (Chang et al., 2022), is used to enhance the model's ability to learn contextual dependencies. Next, the masked features are combined with text features extracted by CLIP (Radford et al., 2021) and fed into the Inter-M Transformer (Javed et al., 2025). Through a spatio-temporal attention mechanism, the model effectively captures spatio-temporal dependencies while aligning text and motion semantics. Finally, a cross-entropy loss is used to predict the masked tokens, facilitating the alignment of cross-modal information.

### 3.3 INFERENCE

As shown in Fig. 3, our architecture starts with a fully masked sequence $t(0)$ and employs the Inter-M Transformer (IMT) to iteratively generate motion tokens for both individuals over $I$ iterations. To enhance the commonsense plausibility, interaction features are incorporated via a residual connection that queries an interaction codebook. At each iteration, the IMT predicts token probabilities at the masked positions. Tokens with the lowest confidence are then resampled and remasked, guiding the model to focus on uncertain regions in subsequent iterations. This iterative refinement continues until the final iteration $I$ is reached. Finally, the tokens are decoded into motion by the decoder.

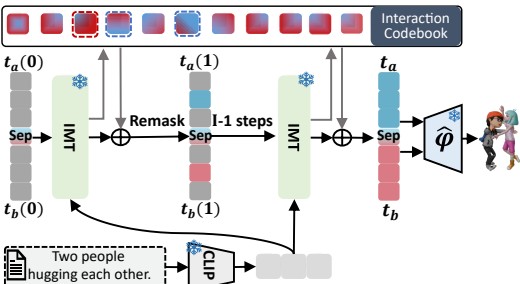

Figure 3: Inference process. Starting from masked Tokens, IMT injects interaction features to produce tokens that are subsequently decoded into motions using the decoder.

## 4 EXPERIMENTS

### 4.1 EXPERIMENTAL SETUP AND EVALUATION

**Dataset.** Following InterMask(Javed et al., 2025), we adopt the InterHuman dataset (Liang et al., 2024) to evaluate CODA on the task of text-conditioned human interaction generation. InterHuman is the first dataset that provides text annotations for two-person interactions, comprising 7,779 motion sequences paired with 23,337 unique descriptions. The dataset is divided into training, validation, and test sets with a ratio of 0.78:0.07:0.15. *More details in Appendix A.*

**Evaluation Metrics.** We quantitatively evaluate the commonsense plausibility of the generated motions from five perspectives. These include realism (*e.g.*, FID), semantic consistency (*e.g.*, R-Precision and MM-Dist), Diversity, physical plausibility (*e.g.*, Physical Body Contact (PBC)), and geometric plausibility (*e.g.*, Interpenetration (IP), Mean Per-Joint Position Error (MPJPE)). *More details in Appendix B.*

**Implementation Details.** For comparison, we adopt the same convolutional residual encoder-decoder architecture as used in InterMask (Javed et al., 2025). The downsampling factor is set to 4, and both the pose and interaction codebooks have a size of 1024. The loss balancing hyper-parameters for the CCL are set as follows: $\lambda_{COM} = 1$, $\lambda_{KTraj} = 1$, $\lambda_{GDM} = 0.5$, and $\lambda_{PPL} = 0.01$. In addition, CLIP-ViT-L/14 (Radford et al., 2021) is used as the text encoder, and the IMT follows the same configuration as in InterMask. During inference, the number of iterations is set to 20 for interaction generation. *More details in Appendix C and D. Our code is in the supplementary materials.*

### 4.2 COMPARE WITH THE STATE-OF-THE-ART

**Comparison of motion quality.** Tab. 1 summarizes the comparison between our method CODA and state-of-the-art approaches on the InterHuman dataset. CODA achieves the best performance in

Table 1: Quantitative evaluation on the InterHuman test set. We run all the evaluations 20 times. ± indicates a 95% confidence interval. Bold indicates the best result, while underline refers to the second best. "→": closer to real motion is better. "†": reproduced results from the official weight.

| Methods | R Precision↑ | | | FID↓ | MM Dist↓ | Diversity→ |
|---|---|---|---|---|---|---|
| | Top 1 | Top 2 | Top 3 | | | |
| Real | $0.452^{\pm.008}$ | $0.610^{\pm.009}$ | $0.701^{\pm.008}$ | $0.273^{\pm.007}$ | $3.755^{\pm.008}$ | $7.948^{\pm.064}$ |
| T2M (Guo et al., 2022) | $0.238^{\pm.012}$ | $0.325^{\pm.010}$ | $0.464^{\pm.014}$ | $13.769^{\pm.072}$ | $4.731^{\pm.013}$ | $7.046^{\pm.022}$ |
| MDM (Tevet et al., 2022) | $0.153^{\pm.012}$ | $0.260^{\pm.009}$ | $0.339^{\pm.012}$ | $9.167^{\pm.056}$ | $6.125^{\pm.018}$ | $7.602^{\pm.045}$ |
| ComMDM (Shafir et al., 2023) | $0.223^{\pm.009}$ | $0.334^{\pm.008}$ | $0.466^{\pm.010}$ | $7.069^{\pm.054}$ | $5.212^{\pm.021}$ | $7.244^{\pm.038}$ |
| FreeMotion (Fan et al., 2025) | $0.326^{\pm.003}$ | $0.462^{\pm.006}$ | $0.544^{\pm.006}$ | $6.740^{\pm.130}$ | $3.848^{\pm.002}$ | $7.828^{\pm.130}$ |
| InterGen† (Liang et al., 2024) | $\underline{0.434}^{\pm.007}$ | $0.592^{\pm.007}$ | $0.672^{\pm.006}$ | $6.446^{\pm.089}$ | $3.797^{\pm.001}$ | $7.872^{\pm.023}$ |
| InterMask† (Javed et al., 2025) | $\mathbf{0.456}^{\pm.004}$ | $\underline{0.610}^{\pm.004}$ | $\underline{0.689}^{\pm.004}$ | $\underline{5.843}^{\pm.088}$ | $\underline{3.792}^{\pm.001}$ | $\underline{8.055}^{\pm.035}$ |
| Ours | $\mathbf{0.456}^{\pm.006}$ | $\mathbf{0.611}^{\pm.005}$ | $\mathbf{0.691}^{\pm.005}$ | $\mathbf{5.358}^{\pm.070}$ | $\mathbf{3.790}^{\pm.001}$ | $\mathbf{8.000}^{\pm.032}$ |

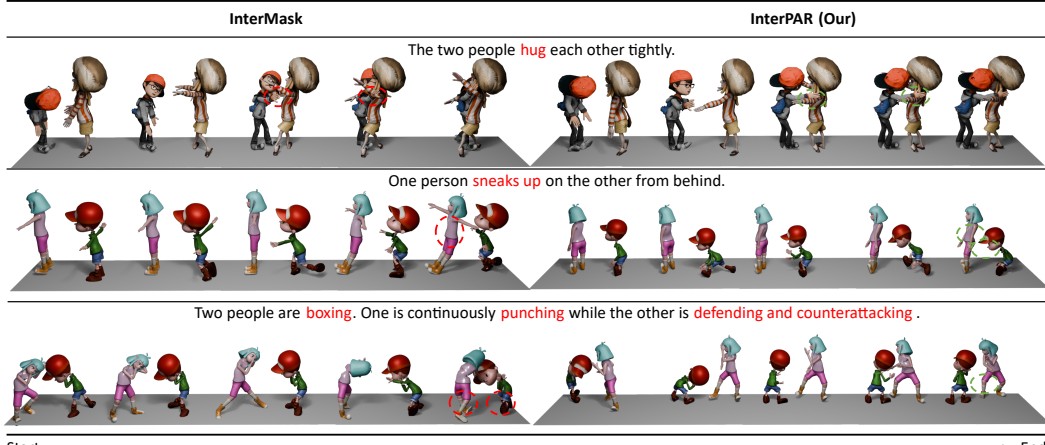

Figure 4: Qualitative comparison between CODA and InterMask on the InterHuman dataset.

R-Precision, FID, MM-Dist, and Diversity. Specifically, compared with diffusion-based generation methods (*e.g.*, FreeMotion (Fan et al., 2025), InterGen (Liang et al., 2024)), CODA increases R Top-3 by **2.1**% and reduces FID by **1.088**, demonstrating its superior ability to maintain text-motion consistency while improving generation quality. Compared with the autoregressive model (*e.g.*, InterMask(Javed et al., 2025)), our approach improves R Top-3 by **0.2**% and lowers FID by **0.485**. These results indicate that the proposed commonsense information storage mechanism and plausible constraints loss effectively suppress motion artifacts and enhance motion quality.

**Comparison of motion plausibility.** Tab.2 compares CODA with state-of-art methods (*e.g.* InterMask) in terms of motion plausibility. Our CODA achieves **-0.003** IP, **-0.004** MPJPE, and the best PBC, indicating more plausible motion generation. These results are attributed to the hierarchical constraints imposed by CCL, which effectively incorporate commonsense priors to mitigate issues such as bone stretching, body leaning, missing contacts, and interpenetration, thereby enhancing the plausibility of the generated motions.

Table 2: Comparison of motion plausibility on the InterHuman dataset.

| Methods | PBC→ | IP↓ | MPJPE↓ |
|---|---|---|---|
| Real | $9.524^{\pm.065}$ | $20.0816^{\pm.022}$ | $0.000^{\pm.000}$ |
| InterMask† | $8.359^{\pm.188}$ | $20.196^{\pm.020}$ | $0.419^{\pm.002}$ |
| Ours | $\mathbf{10.334}^{\pm.220}$ | $\mathbf{20.193}^{\pm.021}$ | $\mathbf{0.415}^{\pm.003}$ |

**Qualitative Comparison.** Fig. 4 provides a qualitative comparison of interaction motion generated by our CODA and InterMask (Javed et al., 2025) on the InterHuman dataset. For the first prompt, InterMask suffers from issues of unintentional separation and penetration, while CODA ensures contact is maintained while avoiding penetration. For the second prompt, InterMask generates a leaning backward issue for the first person, whereas CODA suppresses this unrealistic posture. Finally, for the third prompt, InterMask encounters a skeletal stretching issue, while CODA generates a more reasonable skeletal distribution. These examples demonstrate that CODA generates more realistic, higher-quality, and plausible interactions than InterMask.

## 4.3 ABLATION STUDY

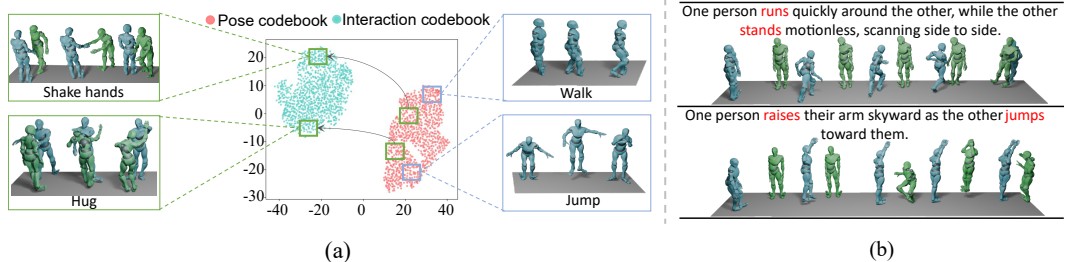

(a)  (b)

Figure 5: (a) t-SNE visualization of codebook features in the ICS; (b) Challenging cases visualization.

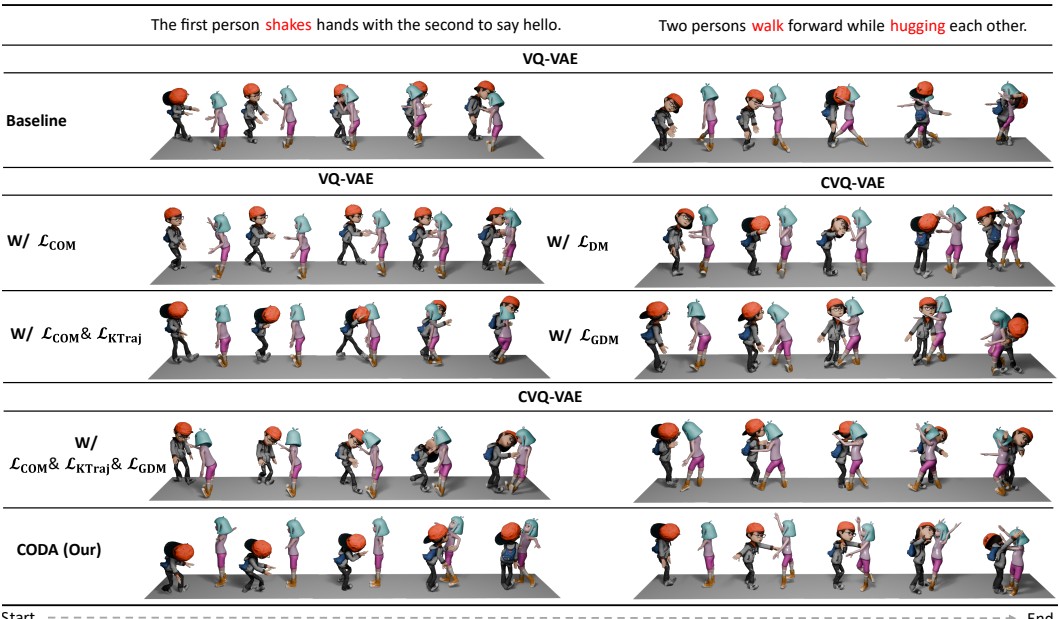

Figure 6: Visual comparison of ablation experiments on the InterHuman dataset.

**Effectiveness of ICS.** Tab.3 and Fig.5(a) provide ablation studies for ICS from both quantitative and qualitative perspectives. In Tab.3, CVQ-VAE *vs* VQ-VAE obtains **-0.24** FID. In Fig.5(a), we use the t-SNE to visualize the distributions of the pose and interaction codebooks

Table 3: Ablation Analysis of VQ-VAE (*i.e.*, Inter-Mask) vs CVQ-VAE (*i.e.*, w/ ICS).

| Methods | Top 1↑ | FID↓ | MM Dist↓ | Diversity→ |
|---|---|---|---|---|
| VQ-VAE | **0.456**$^{\pm.004}$ | 5.843$^{\pm.088}$ | 3.792$^{\pm.001}$ | 8.055$^{\pm.035}$ |
| CVQ-VAE | 0.448$^{\pm.006}$ | **5.603**$^{\pm.064}$ | **3.791**$^{\pm.001}$ | **8.006**$^{\pm.032}$ |

while also locally visualizing their corresponding motions. We observe that two codebooks form distinct distributions, with each codebook capturing its respective motion patterns. These results prove that ICS has an excellent ability to store single- and multi-person motion information, which provides a foundation for enhancing motion plausibility. *More details in Appendix I.*

**Effectiveness of Single-Person loss constraints.** Tab. 4 presents an ablation study of our proposed single-person loss constraints $\mathcal{L}_{\text{COM}}$ and $\mathcal{L}_{\text{KTraj}}$. We can observe that incorporating $\mathcal{L}_{\text{COM}}$ significantly reduces the FID score by **0.514** (A *vs* B), while introducing $\mathcal{L}_{\text{KTraj}}$ increases multimodality by **0.044** (A *vs* C). In Fig. 6, the motion generated by InterMask exhibits issues such as leaning back and skeletal stretching. In contrast, incorporating the single-person loss constraints into InterMask results in correct skeletal motion during the *"shake"* action, without leaning backward problems. These results show that the $\mathcal{L}_{\text{COM}}$ improves body balance, while the $\mathcal{L}_{\text{KTraj}}$ constrains excessive bone stretching. *More experiments in Appendix H.*

**Effectiveness of Human-Human loss constraints.** Tab. 5, Tab. 6, and Fig. 6 evaluate the effectiveness of Human-Human loss constraints (*e.g.*, $\mathcal{L}_{\text{GDM}}$ and $\mathcal{L}_{\text{PPL}}$) in mitigating penetration and non-contact issues. In Tab. 5, starting from a COM loss baseline, adding the $\mathcal{L}_{\text{GDM}}$ reduces the FID by **0.1** ($\mathcal{L}_{\text{GDM}}$ *vs* $\mathcal{L}_{\text{DM}}$ (Liang et al., 2024)), indicating effective alleviation of non-contact problems. This improvement is attributed to the soft-thresholding strategy, which grants the model greater

Table 4: Ablation study on single-person constraint losses of CODA (VQ-VAE) on the InterHuman test dataset. Baseline denotes InterMask.

| Index | $\mathcal{L}_{COM}$ | $\mathcal{L}_{KTraj}$ | TOP 1 ↑ | FID ↓ | Diversity → |
|---|---|---|---|---|---|
| A | Baseline | | $\mathbf{0.456}^{\pm.004}$ | $\underline{5.843}^{\pm.088}$ | $8.055^{\pm.035}$ |
| B | ✓ | | $\underline{0.450}^{\pm.005}$ | $\mathbf{5.043}^{\pm.086}$ | $7.996^{\pm.023}$ |
| C | ✓ | ✓ | $0.440^{\pm.005}$ | $5.846^{\pm.091}$ | $\mathbf{8.040}^{\pm.032}$ |

Table 5: Ablation study on contact loss function of CODA (CVQ-VAE). $\mathcal{L}_{DM}$ represents the masked joint distance map loss.

| Index | $\mathcal{L}_{DM}$ | $\mathcal{L}_{GDM}$ | TOP 1 ↑ | FID ↓ | Diversity → |
|---|---|---|---|---|---|
| A | ✓ | | $\mathbf{0.464}^{\pm.006}$ | $5.487^{\pm.057}$ | $7.967^{\pm.030}$ |
| B | | ✓ | $0.443^{\pm.007}$ | $\mathbf{5.387}^{\pm.071}$ | $\mathbf{0.767}^{\pm.029}$ |

Table 6: Ablation study on Human-Human constraint losses of CODA on the InterHuman dataset.

| Index | $\mathcal{L}_{COM}$ | $\mathcal{L}_{KTraj}$ | $\mathcal{L}_{GDM}$ | $\mathcal{L}_{PPL}$ | TOP 1 ↑ | FID ↓ | MM Dist ↓ | Diversity → |
|---|---|---|---|---|---|---|---|---|
| | | | | | **CVQ-VAE** | | | |
| A | ✓ | ✓ | ✓ | | $0.443^{\pm.005}$ | $5.979^{\pm.086}$ | $3.796^{\pm.001}$ | $8.033^{\pm.036}$ |
| B | ✓ | ✓ | ✓ | ✓ | $\mathbf{0.456}^{\pm.006}$ | $\mathbf{5.358}^{\pm.070}$ | $\mathbf{3.790}^{\pm.001}$ | $\mathbf{8.000}^{\pm.032}$ |

flexibility in capturing trends in contact motions. In Tab. 6, while KTraj loss slightly enhances diversity, it leads to a decline in motion generation quality. In contrast, incorporating our proposed $\mathcal{L}_{PPL}$ substantially improves all evaluated metrics. demonstrating its ability to balance multiple loss objectives. Furthermore, Fig. 6 illustrates that generation methods without loss constraints suffer from non-contact and penetration artifacts. By contrast, motions generated with Human-Human loss constraints successfully capture interactive actions such as "*shake*" and "*hug*". These results demonstrate that the commonsense constraints introduced by CODA effectively enhance the plausibility of behavior generation, successfully mitigating issues such as skeletal stretching, body leaning, non-contact, and penetration. *For more details, please refer to the supplementary video.*

**Visualization of Challenging Cases.** To verify the robustness of CODA, we performed experiments involving intense movements such as running and jumping. In Fig. 5 (b), CODA is capable of performing the first instruction "*run*" while simultaneously executing the second instruction "*jump*" with mid-air actions. These success cases are attributed to: (1) the ICS and large code size enhance the richness of motion information and ensure higher motion quality; (2) Jointly employing $\mathcal{L}_{COM}$ and $\mathcal{L}_{KTraj}$ improves body stability and constrains excessive skeletal stretching.

**Plug-and-play validation of CCL.** Tab. 7 presents ablation studies by adding $\mathcal{L}_{CCL}$ to InterGen and InterMask to verify its plug-and-play property. We observe that after adding $\mathcal{L}_{CCL}$ to InterGen, R Top1 increases by **5.7**% and FID decreases by **0.674**. For InterMask, adding this loss keeps R Top1 unchanged while reducing FID by **0.485**. Both models show significant performance improvements, demonstrating that CCL can not only be seamlessly applied to diffusion and autoregressive methods but also enhances text-to-motion consistency and generates motion quality.

Table 7: Plug-and-play experiment of CCL.

| Methods | Top 1 ↑ | FID ↓ | MM Dist ↓ | Diversity → |
|---|---|---|---|---|
| InterGen | $0.434^{\pm.007}$ | $6.446^{\pm.089}$ | $3.797^{\pm.001}$ | $7.872^{\pm.023}$ |
| w/ $\mathcal{L}_{CCL}$ | $\mathbf{0.491}^{\pm.007}$ | $\mathbf{5.762}^{\pm.079}$ | $\mathbf{3.772}^{\pm.001}$ | $7.891^{\pm.025}$ |
| InterMask | $\mathbf{0.456}^{\pm.004}$ | $\underline{5.843}^{\pm.088}$ | $3.792^{\pm.001}$ | $8.055^{\pm.035}$ |
| w/ $\mathcal{L}_{CCL}$ | $\mathbf{0.456}^{\pm.006}$ | $\mathbf{5.358}^{\pm.070}$ | $\mathbf{3.790}^{\pm.001}$ | $8.000^{\pm.032}$ |

## 5 CONCLUSION AND LIMITATION

**Conclusion.** In this work, we proposed CODA, a novel framework for human interaction generation, which achieves commonsense plausibility without relying on motion simulators. By leveraging ICS as a commonsense memory and CCL as a hierarchical constraint, CODA effectively suppresses motion artifacts such as bone stretching, body leaning, missing contacts, and interpenetration. Extensive experiments demonstrate that CODA is efficient, flexible, and outperforms state-of-the-art methods in generating high-quality, commonsense-plausible human motions.

**Limitations and Future Work.** While our method significantly improves the commonsense plausibility of human interaction generation, it is currently limited by its training on single- and two-person datasets, which restricts its direct applicability to more complex multi-person interaction scenarios. To overcome this limitation, future work will explore enhancing the cross-attention mechanism with a multi-codebook storage strategy, enabling the concatenation and joint modeling of features from multiple individuals. This extension is expected to facilitate more natural, coherent, and scalable modeling of complex multi-person interactions.

ETHICS STATEMENT

This work does not involve sensitive personal data or applications with direct societal risks. All datasets used are publicly available and have been widely adopted in prior research. We therefore believe our study poses no ethical concerns beyond standard practices.

REPRODUCIBILITY STATEMENT

To facilitate reproducibility, we provide detailed descriptions of dataset usage and model hyper-parameters in Appendix A–F. All datasets are publicly available, and the code is included in the supplemental materials.

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

**LLM usage statement**. We employed large language models (LLMs) as auxiliary tools during manuscript preparation. Their use was strictly limited to surface-level editing tasks, including grammar correction, minor rephrasing, and stylistic improvements to enhance readability. At no point did we rely on LLMs for generating research ideas, methods, experiments, or conclusions. All technical content and analysis presented in this paper are the sole work of the authors.

**Overview.** The supplementary includes the following sections:

- Human pose data and rendering methods (§A).
- Evaluation metrics (§B).
- Implementation Details (§C and §D).
- Algorithm (§E).
- Hyperparameter Setup (§F).
- More Our Results (§G).
- More Experiments on KIT-ML (§H).
- Visualization Details of ICS(§I).
- Potential Broader Impacts (§J).

**The visualization videos and code will be provided in the supplementary materials package.**

## A  HUMAN POSE DATA AND RENDERING METHODS

In this section, we first introduce the skeletal data of the InterHuman dataset, followed by the skinning method and rendering tools employed. The specific details are presented as follows:

**InterHuman dataset.** It is the first dataset with text annotations for two-person interactions, containing 7,779 motion sequences and 23,337 unique descriptions. As shown in Fig. 7 (a), InterHuman follows the AMASS (Mahmood et al., 2019) skeleton representation with 22 joints, including the root joint. Each motion sequence representation is formulated as:

$$x^i = [\mathbf{j}_g^p, \mathbf{j}_g^v, \mathbf{j}^r, \mathbf{c}^f] \tag{10}$$

where the $i$-th motion state $x^i$ is defined as a collection of global joint positions $\mathbf{j}_g^p \in \mathbb{R}^{3N_j}$, velocities $\mathbf{j}_g^v \in \mathbb{R}^{3N_j}$ in the world frame, 6D representation of local rotations $\mathbf{j}^r \in \mathbb{R}^{6N_j}$ in the root frame, and binary foot-ground contact features $\mathbf{c}^f \in \mathbb{R}^4$, $N_j = 22$ denotes the joints number.

**Skinning and rendering.** To render the generated human skeletons in an aesthetically pleasing anime style, we first downloaded the T-Pose skinned models of anime characters from Mixamo in Fig. 7 (b). Next, the generated human skeletons were converted into BioVision Hierarchy (BVH) format compatible with Blender. Finally, using the Rokoko plugin in Blender 4.2, the BVH skeleton animations were rigged to the T-Pose skinned models according to the skeletal joints, thereby achieving synchronized motion between the skeletons and the anime characters.

## B  EVALUATION METRICS

We quantitatively evaluate the commonsense plausibility of the generated motions from five perspectives. These include realism (*e.g.*, FID), semantic consistency (*e.g.*, R-Precision and MM-Dist), Diversity, physical plausibility (*e.g.*, Physical Body Contact (PBC)), and geometric plausibility (*e.g.*, Interpenetration (IP), Mean Per-Joint Position Error (MPJPE)). The details are as follows:

**Fréchet Inception Distance (FID)**: FID is computed by first extracting features from both generated and real motions. Then it evaluates the distance between the distributions of these two feature sets.

**R-Precision**: For each generated motion, a candidate pool is constructed comprising its ground-truth textual description and 31 randomly selected mismatched descriptions from the test set. The Euclidean distances between the motion feature and the text features of all candidates are computed and ranked. Retrieval is considered successful if the ground-truth description appears in the top-$k$ entries. The final R-Precision is reported as the average accuracy at top-1, top-2, and top-3 ranks.

Figure 7: (a) AMSS skeleton; (b) The character skinning from the Mixamo website.

**Matching Distance (MM Dist)**: MM Distance quantifies the alignment between generated motions and their corresponding text descriptions. It is defined as the mean Euclidean distance between the motion feature and the feature of its associated text description in the test set.

**Diversity**: Diversity measures the variability among generated motions. Two random subsets of generated motions, each of size $S_d$, are sampled. For each subset, motion feature vectors $\{\mathbf{v}_1, \ldots, \mathbf{v}_{S_d}\}$ and $\{\mathbf{v}'_1, \ldots, \mathbf{v}'_{S_d}\}$ are extracted. Diversity is then computed as:

$$\text{Diversity} = \frac{1}{S_d} \sum_{i=1}^{S_d} \|\mathbf{v}_i - \mathbf{v}'_i\|_2 . \tag{11}$$

**Physical Body Contact (PBC)**: To overcome the limitation of the Physical Foot Contact Score, which only considers the lower body, PBC (Physical Body Contact) incorporates factors related to the neck and hands based on PFC, extending the evaluation scope to the entire body (Luo et al., 2024). The formulas are as follows:

$$\text{PBC} = \frac{1}{N} \sum_{i=1}^{N} \left[ -v_{\text{lfoot}}^i \cdot v_{\text{rfoot}}^i \cdot a_{\text{root}}^i + v_{\text{lhand}}^i \cdot v_{\text{lchest}}^i \cdot a_{\text{lchest}}^i + v_{\text{rhand}}^i \cdot v_{\text{rchest}}^i \cdot a_{\text{rchest}}^i + v_{\text{head}}^i \cdot v_{\text{neck}}^i \cdot a_{\text{neck}}^i \right] \tag{12}$$

where "l" denotes the joints on the left side and "r" denotes the joints on the right side.

**Interpenetration**: Interpenetration computes the average volumetric overlap of human meshes in two-person scenarios. Specifically, each human body is approximated by 45 spheres, and the volume intersecting these spheres is calculated to obtain a measure of interpenetration (Yao et al., 2025).

**Mean Per-Joint Position Error (MPJPE)**: MPJPE compares joint positions between position-based and rotation-based motion representations. It captures discrepancies between motion components, emphasizing the importance of decoupled evaluation and application.

## C  BASELINE IMPLEMENTATION

To ensure fair comparisons in our experiments, all baseline methods are trained and tested using a two-stage process on the same dataset. During training, the first stage employs a VQ-VAE model with a batch size of 512. In the second stage, due to the memory limitations of the RTX 3090 GPU, we adopt a gradient accumulation strategy in the Inter-M Transformer to mitigate performance degradation caused by a smaller batch size. Specifically, the batch size is set to 12 with 8 accumulation

steps. Except for the above settings, all other training and inference hyperparameters follow the official configurations provided in the InterMask (Javed et al., 2025).

# D  IMPLEMENTATION DETAILS

In this section, we describe the proposed CODA model architecture and the experimental details of the training and inference stages. Further details are provided in Section D.1 and Section D.2.

## D.1  MODEL ARCHITECTURE

The Motion CVQ-VAE architecture utilizes 2D convolutional residual blocks in both its encoder and decoder. Temporal downsampling is fixed at a ratio of $n/N = 1/4$ across both datasets, while spatial downsampling is dataset-specific: $j/J = 5/22$ for InterHuman. Downsampling in the encoder is achieved using strided convolutions, whereas the decoder employs upsampling followed by convolutional layers to restore the original dimensions. The latent representations generated by the CVQ-VAE have a dimensionality of $d' = 512$. The size of the single-person pose codebook is $|\mathcal{C}| = 1024$, and the size of the interaction codebook is $|\rho| = 1024$.

For the Inter-M transformer, we adopt $\mathbf{L} = 6$ transformer blocks, each comprising 6 attention heads. The embedding dimension for the transformer is set to $\tilde{d} = 384$.

Table 8: CVQ-VAE and Inter-M Transformer Model Parameters.

| Parameter | Value | Description |
|---|---|---|
| $d'$ | 512 | Latent space dimension of CVQ-VAE |
| $|\mathcal{C}|$ | 1024 | Single-person pose codebook size |
| $|\rho|$ | 1024 | Interaction Codebook size |
| $n/N$ | 1/4 | Temporal downsampling factor for both datasets |
| $j/J$ (InterHuman) | 5/22 | Spatial downsampling for InterHuman dataset |
| $\mathbf{L}$ | 6 | Number of transformer blocks |
| Attention heads | 6 | Number of attention heads per block |
| $\tilde{d}$ | 384 | Transformer embedding dimension |
| CLIP version | ViT-L/14@336px | Version of CLIP used for text in transformer |

Table 9: Training Hyperparameters for the CVQ-VAE and Inter-M Transformer.

| Parameter | Value | Description |
|---|---|---|
| CVQ-VAE batch size | 256*2 | Batch size and accumulation steps of CVQ-VAE |
| Transformer batch size | 12*8 | Batch size and accumulation steps of transformer |
| Initial learning rate | 0.0002 | Starting learning rate for both models |
| Learning rate decay | 0.1 / 1/3 | Decay factor for CVQ-VAE / Transformer learning rate |
| $\alpha$ and $\beta$ | 0.02 | Commitment loss factor for CVQ-VAE |
| $\lambda_{vel}, \lambda_{fc}, \lambda_{bl}$ (InterHuman) | 100, 500, 5 | Geometric loss weights for InterHuman |
| Condition drop prob. | 0.1 | Drop probability for text conditioning during transformer training |
| $p_r$ | 0.8 | Random Masking probability for stage 1 masking during training |

## D.2  TRAINING AND INFERENCE DETAILS

The CVQ-VAE is trained for 50 epochs using a batch size of 256 with gradient accumulation, where the accumulation step is set to 2. The learning rate is initialized at 0.0002 and follows a multistep decay schedule, decreasing by a factor of 0.1 after 70% and 85% of the total iterations. A linear warm-up is applied during the first 25% of the iterations. The geometric losses for velocity, foot contact, and bone length are weighted differently in the data sets.

The Inter-M transformer is trained for 500 epochs using gradient accumulation with a batch size of 12 and an accumulation step of 8. A similar multistep learning rate decay strategy is employed, where the learning rate is reduced by a factor of 1/3 at 50%, 70%, and 85% of the total iterations.

During inference, the number of iterations $I$ is set to 20 for interaction generation. A classifier-free guidance (CFG) scale of 2 is applied, and the temperature is set to 1 to balance diversity and coherence in the generated results.

# E    ALGORITHM

Algorithm 1 provides a detailed computation process of the physical constraint loss. To ensure reproducibility, we will release the code in the future.

---

**Algorithm 1** The computation of Physical Constraint Loss (CCL)

---

**Require:** $\hat{\mathbf{m}}$, $\mathbf{m}$: Predicted and ground-truth motions with shape $[B, T, J, 3]$, $\lambda_{\text{COM}}$, $\lambda_{\text{KTraj}}$, $\lambda_{\text{GDM}}$, and $\lambda_{\text{PPL}}$: Weight coefficients, $\tau$: Temperature coefficient, CH: Number of body-part chains, root: Pelvis joint.

**Ensure:** $\mathcal{L}_{\text{CCL}}$: Commonsense Constraint Loss

1: /* Common Geometric Loss (*e.g.*, foot contact, velocity, bone length)*/
2: $\mathcal{L}_{\text{geo}} \leftarrow \mathcal{L}_{\text{fc}} + \mathcal{L}_{\text{vel}} + \mathcal{L}_{\text{bl}}$

3: /* Center-of-Mass (COM) Loss*/
4: $\hat{c} \leftarrow \frac{1}{J} \sum_j \hat{\mathbf{m}}_j, \quad c \leftarrow \frac{1}{J} \sum_j \mathbf{m}_j$
5: $\texttt{C}(\cdot) \leftarrow \|c^* - \text{root}\|_2$
6: $\texttt{A}(\cdot) \leftarrow \cos(c^* - \text{root}, \text{ XY-plane})$
7: $\mathcal{L}_{\text{COM}} \leftarrow \|\texttt{C}(\hat{\mathbf{m}}) - \texttt{C}(\mathbf{m})\|_2^2 + \|\texttt{A}(\hat{\mathbf{m}}) - \texttt{A}(\mathbf{m})\|_2^2$

8: /* Key Joint Trajectory Loss */
9: $k \leftarrow$ Select key joints (*e.g.*, hands, feet, pelvis).
10: $\mathcal{L}_{\text{KTraj}} \leftarrow \frac{1}{T} \sum_t \|\hat{\mathbf{m}}_k(t) - \mathbf{m}_k(t)\|_1$

11: /* Gaussian Joint Distance Map (GDM) Loss */
12: $D_{\text{pred}}, D_{\text{tgt}} \leftarrow$ Compute pairwise distance maps
13: $W_{\text{pred}} \leftarrow \exp\left(-\frac{(D_{\text{pred}})^2}{2\tau^2}\right), \quad W_{\text{tgt}} \leftarrow \exp\left(-\frac{(D_{\text{tgt}})^2}{2\tau^2}\right)$
14: $\mathcal{L}_{\text{GDM}}^{(1)} \leftarrow \frac{|D_{\text{pred}} - D_{\text{tgt}}| \cdot W_{\text{pred}}}{\sum W_{\text{pred}} + \varepsilon}, \quad \mathcal{L}_{\text{GDM}}^{(2)} \leftarrow \frac{|D_{\text{pred}}| \cdot W_{\text{tgt}}}{\sum W_{\text{tgt}} + \varepsilon}$
15: $\mathcal{L}_{\text{GDM}} \leftarrow \mathcal{L}_{\text{GDM}}^{(1)} + \mathcal{L}_{\text{GDM}}^{(2)}$

16: /* Penetration Penalty Loss (PPL) */
17: $\mathcal{L}_{\text{PPL}} \leftarrow 0$
18: **for** ch = 1 to CH **do**
19:      $\mathbb{1}_{\text{overlap}}^{\text{ch}} \leftarrow$ Axis-Aligned Bounding Boxes (AABB) overlap
20:      $\mathbf{d}^{\text{pred}}, \mathbf{d}^{\text{gt}} \leftarrow$ Minimum inter-chain distance
21:      $\mathcal{L}_{\text{PPL}}^{\text{ch}} \leftarrow \mathbb{1}_{\text{overlap}}^{\text{ch}} \times \left[\max(0, \ 0.9 \times \mathbf{d}^{\text{gt}} - \mathbf{d}^{\text{pred}}) + (\mathbf{d}^{\text{pred}} - \mathbf{d}^{\text{gt}})^2\right]$
22:      $\mathcal{L}_{\text{PPL}} \leftarrow \mathcal{L}_{\text{PPL}} + \mathcal{L}_{\text{PPL}}^{\text{ch}}$
23: **end for**
24: $\mathcal{L}_{\text{PPL}} \leftarrow \frac{1}{CH} \mathcal{L}_{\text{PPL}}$

25: /* Final Aggregation */
26: $\mathcal{L}_{\text{CCL}} \leftarrow \mathcal{L}_{\text{geo}} + \lambda_{\text{COM}} \mathcal{L}_{\text{COM}} + \lambda_{\text{KTraj}} \mathcal{L}_{\text{KTraj}} + \lambda_{\text{GDM}} \mathcal{L}_{\text{GDM}} + \lambda_{\text{PPL}} \mathcal{L}_{\text{PPL}}$
27: **return** $\mathcal{L}_{\text{CCL}}$

---

# F    HYPERPARAMETER SETUP

As shown in Tab. 10–13 and Fig.8, we systematically evaluated the impact of four physical constraint loss hyperparameters (*e.g.*, $\lambda_{COM}$, $\lambda_{KTraj}$, $\lambda_{GDM}$, *and* $\lambda_{PPL}$) on model performance. The experimental results indicate that both excessively large and small values of these hyperparameters negatively affect generation quality and motion consistency. Specifically, when the weights are too small, the physical constraints are insufficient, causing unrealistic behaviors such as backward leaning, abnormal bone stretching, lack of ground contact, and bone penetration. Conversely, when the weights are too large,

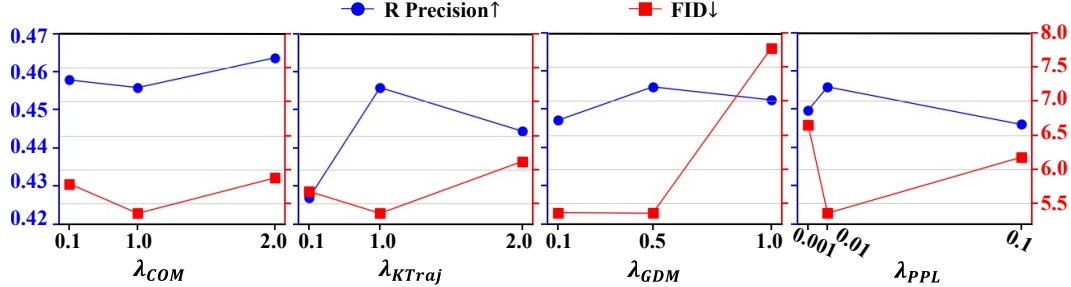

Figure 8: Ablation analysis of hyperparameter settings on InterHuman test dataset. R Precision measures text-to-action consistency, while FID evaluates the quality of generated motions.

Table 10: Effect of $\lambda_{\text{COM}}$ on performance metrics.

| $\lambda_{\text{COM}}$ | R Precision↑ | | | FID↓ | MM Dist↓ | Diversity→ |
|---|---|---|---|---|---|---|
| | Top 1 | Top 2 | Top 3 | | | |
| 0.5 | $0.458^{\pm.007}$ | $0.615^{\pm.006}$ | $0.696^{\pm.004}$ | $5.792^{\pm.074}$ | $3.787^{\pm.001}$ | $8.004^{\pm.031}$ |
| 1 | $0.456^{\pm.006}$ | $0.611^{\pm.005}$ | $0.691^{\pm.005}$ | $\mathbf{5.358}^{\pm.070}$ | $3.790^{\pm.001}$ | $\mathbf{8.000}^{\pm.032}$ |
| 2 | $\mathbf{0.464}^{\pm.005}$ | $\mathbf{0.621}^{\pm.006}$ | $\mathbf{0.704}^{\pm.005}$ | $5.877^{\pm.073}$ | $\mathbf{3.786}^{\pm.001}$ | $8.004^{\pm.032}$ |

Table 11: Effect of $\lambda_{\text{KTraj}}$ on performance metrics.

| $\lambda_{\text{KTraj}}$ | R Precision↑ | | | FID↓ | MM Dist↓ | Diversity→ |
|---|---|---|---|---|---|---|
| | Top 1 | Top 2 | Top 3 | | | |
| 0.5 | $0.427^{\pm.006}$ | $0.587^{\pm.006}$ | $0.672^{\pm.005}$ | $5.680^{\pm.089}$ | $3.797^{\pm.001}$ | $7.890^{\pm.033}$ |
| 1 | $\mathbf{0.456}^{\pm.006}$ | $\mathbf{0.611}^{\pm.005}$ | $\mathbf{0.691}^{\pm.005}$ | $\mathbf{5.358}^{\pm.070}$ | $\mathbf{3.790}^{\pm.001}$ | $8.000^{\pm.032}$ |
| 2 | $0.444^{\pm.005}$ | $0.602^{\pm.006}$ | $0.685^{\pm.005}$ | $6.119^{\pm.074}$ | $3.795^{\pm.001}$ | $\mathbf{7.975}^{\pm.031}$ |

Table 12: Effect of $\lambda_{\text{GDM}}$ on performance metrics.

| $\lambda_{\text{GDM}}$ | R Precision↑ | | | FID↓ | MM Dist↓ | Diversity→ |
|---|---|---|---|---|---|---|
| | Top 1 | Top 2 | Top 3 | | | |
| 0.1 | $0.447^{\pm.005}$ | $0.598^{\pm.004}$ | $0.680^{\pm.005}$ | $5.367^{\pm.059}$ | $3.792^{\pm.001}$ | $\mathbf{7.916}^{\pm.042}$ |
| 0.5 | $\mathbf{0.456}^{\pm.006}$ | $\mathbf{0.611}^{\pm.005}$ | $\mathbf{0.691}^{\pm.005}$ | $\mathbf{5.358}^{\pm.070}$ | $\mathbf{3.790}^{\pm.001}$ | $8.000^{\pm.032}$ |
| 1 | $0.452^{\pm.006}$ | $0.608^{\pm.006}$ | $0.690^{\pm.005}$ | $7.767^{\pm.113}$ | $3.796^{\pm.001}$ | $8.077^{\pm.030}$ |

the physical loss terms dominate the optimization process, overly restricting the model's flexibility and resulting in overly conservative generations that degrade overall quality and diversity. Only when $\lambda_{\text{COM}} = 1$, $\lambda_{\text{KTraj}} = 1$, $\lambda_{\text{GDM}} = 0.5$, and $\lambda_{\text{PPL}} = 0.01$ do the loss terms achieve a good balance and synergy, effectively guiding the generation process to ensure physical plausibility while improving generation quality and motion consistency.

To verify the impact of the size of the codebook on the quality of motion generation, we set the size of the codebook to 128, 256, 512, and 1024. As shown in Tab. 14, FID decreases consistently as the size increases, indicating that a larger size enhances the richness of stored information and effectively improves motion generation quality. The fluctuations in R Precision and MM Dist suggest that only when the size is sufficiently large can the information be rich enough to ensure reliable text–motion consistency.

Table 13: Effect of $\lambda_{\text{PPL}}$ on performance metrics.

| $\lambda_{\text{PPL}}$ | R Precision↑ | | | FID↓ | MM Dist↓ | Diversity→ |
|---|---|---|---|---|---|---|
| | Top 1 | Top 2 | Top 3 | | | |
| 0.001 | $\underline{0.450}^{\pm.005}$ | $\underline{0.607}^{\pm.006}$ | $\underline{0.688}^{\pm.006}$ | $6.649^{\pm.097}$ | $3.794^{\pm.001}$ | $\underline{8.045}^{\pm.036}$ |
| 0.01 | $\mathbf{0.456}^{\pm.006}$ | $\mathbf{0.611}^{\pm.005}$ | $\mathbf{0.691}^{\pm.005}$ | $\mathbf{5.358}^{\pm.070}$ | $\mathbf{3.790}^{\pm.001}$ | $\mathbf{8.000}^{\pm.032}$ |
| 0.1 | $0.446^{\pm.006}$ | $0.604^{\pm.005}$ | $0.684^{\pm.005}$ | $\underline{6.175}^{\pm.111}$ | $\underline{3.794}^{\pm.001}$ | $7.861^{\pm.035}$ |

Table 14: Effect of CodeSize on performance metrics.

| CodeSize | R Precision↑ | | | FID↓ | MM Dist↓ | Diversity→ |
|---|---|---|---|---|---|---|
| | Top 1 | Top 2 | Top 3 | | | |
| 128 | $0.437^{\pm.006}$ | $0.590^{\pm.005}$ | $0.671^{\pm.005}$ | $8.813^{\pm.094}$ | $3.801^{\pm.001}$ | $7.845^{\pm.026}$ |
| 256 | $0.427^{\pm.004}$ | $0.578^{\pm.004}$ | $0.661^{\pm.005}$ | $6.898^{\pm.065}$ | $3.804^{\pm.001}$ | $\mathbf{7.953}^{\pm.030}$ |
| 512 | $0.397^{\pm.006}$ | $0.558^{\pm.006}$ | $0.647^{\pm.005}$ | $6.288^{\pm.088}$ | $3.811^{\pm.001}$ | $8.055^{\pm.035}$ |
| 1024 | $\mathbf{0.456}^{\pm.006}$ | $\mathbf{0.611}^{\pm.005}$ | $\mathbf{0.691}^{\pm.005}$ | $\mathbf{5.358}^{\pm.070}$ | $\mathbf{3.790}^{\pm.001}$ | $\underline{8.000}^{\pm.032}$ |

## G  MORE OUR RESULTS

In this section, we provide more qualitative results of our CODA. As shown in Fig. 9, we can observe that CODA is capable of generating physically plausible motions (*e.g.*, walking, hugging, and attacking) while maintaining consistency (*e.g.*, waving, sitting, and blaming) between text and motions. This further validates the original intention of the proposed method to enhance the plausibility of motion generation.

## H  MORE EXPERIMENTS ON KIT-ML

**KIT-ML Dataset.** It contains 3,911 motion sequences accompanied by 6,278 text annotations. Each pose is represented by a 251-dimensional feature vector capturing similar global and local motion attributes, with local information extracted from 21 joints aligned to the SMPL model. The KIT-ML dataset is divided into training, validation, and testing sets with a ratio of 0.8:0.05:0.15.

**Quantitative Results.** Tab. 15 reports the comparison results on the single-person motion generation dataset. From the results, we observe that our designed COM loss and KTraj loss reduce the FID by **2.3**%, MM Dist by **5.8**%, and multimodality by **0.117** compared to MoMask (Guo et al., 2024). These results demonstrate that our proposed losses exhibit strong adaptability and effectively improve motion quality and diversity.

## I  VISUALIZATION DETAILS OF ICS

To validate the effectiveness of the ICS, we visualize the pose and interaction codebook indices corresponding to the behaviors in Fig.10. We observe the following: (1) the single-person pose codebook can effectively reconstruct action sequences that align with single-person descriptions; (2) when two pose codebooks are combined with one interaction codebook, the generated two-person action sequences not only maintain motion diversity but also exhibit improved plausibility due to the constraints imposed by the interaction loss; (3) the occurrence of repeated indices in the codebooks is mainly due to the sustained holding of motion poses. These results demonstrate that the proposed ICS method is effective in capturing both single-person and multi-person motion features.

## J  POTENTIAL BROADER IMPACTS

The proposed CODA introduces a novel framework for human-to-human interaction generation, which may involve the following broader impacts:

Table 15: Quantitative evaluation on KIT-ML (Plappert et al., 2016) test set. "→": closer to real motion is better.

| Methods | R Precision↑ | | | FID↓ | MM Dist↓ | Diversity→ |
|---|---|---|---|---|---|---|
| | Top 1 | Top 2 | Top 3 | | | |
| Real | $0.424^{\pm.005}$ | $0.649^{\pm.006}$ | $0.779^{\pm.006}$ | $0.031^{\pm.004}$ | $2.788^{\pm.012}$ | $11.080^{\pm.097}$ |
| T2M-GPT (Zhang et al., 2023) | $0.402^{\pm.006}$ | $0.619^{\pm.005}$ | $0.737^{\pm.006}$ | $0.717^{\pm.041}$ | $3.053^{\pm.026}$ | $10.86^{\pm.094}$ |
| AttT2M (Zhong et al., 2023) | $0.413^{\pm.006}$ | $0.632^{\pm.006}$ | $0.751^{\pm.006}$ | $0.870^{\pm.039}$ | $3.039^{\pm.021}$ | $\underline{10.96}^{\pm.123}$ |
| MMM (Pinyoanuntapong et al., 2024) | $0.404^{\pm.005}$ | $0.621^{\pm.005}$ | $0.744^{\pm.004}$ | $0.316^{\pm.028}$ | $2.977^{\pm.019}$ | $10.91^{\pm.101}$ |
| BAD (Hosseyni et al., 2025) | $0.417^{\pm.006}$ | $0.631^{\pm.005}$ | $0.750^{\pm.005}$ | $0.221^{\pm.012}$ | $2.941^{\pm.025}$ | $\mathbf{11.00}^{\pm.100}$ |
| MoMask (Guo et al., 2024) | $\underline{0.433}^{\pm.007}$ | $\mathbf{0.656}^{\pm.005}$ | $\mathbf{0.781}^{\pm.005}$ | $\underline{0.204}^{\pm.011}$ | $\underline{2.779}^{\pm.022}$ | - |
| Ours | $\mathbf{0.437}^{\pm.006}$ | $\underline{0.653}^{\pm.006}$ | $\underline{0.778}^{\pm.004}$ | $\mathbf{0.181}^{\pm.015}$ | $\mathbf{2.721}^{\pm.018}$ | $10.838^{\pm.083}$ |

One person approaches the other.

Two people are waving their hands and performing a dance step together.

First person is sitting in a chair, the second takes a step forward with their right foot.

The two are blaming each other and having an intense argument.

Two persons walk forward while hugging each other.

Both people are doing fencing practice, attacking each other with their swords. during the practice, the first person make a short lunge and touches the tip of the sword to the top of the second's head.

Start ----------------------------------------→ End

Figure 9: More qualitative results of CODA on the task of text-to-motion.

- **Enhanced Human-Robot Collaboration Safety.** CODA improves human-robot interaction by generating contextually appropriate and realistic human motions. In shared workspaces or service environments, accurately generated human motion sequences help robots anticipate human motions, thereby reducing the risk of collisions or unsafe responses.

- **More Immersive VR, AR, and Gaming Experiences.** By generating lifelike human motions in interactive scenarios, CODA can enrich virtual characters' responsiveness and

One person walk forward.

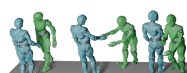

Pose codebobok Index

[224,374,374,374,374,374,83,293,67,67,83,485,809,89,161,233,693,693,238,43,630,271,875,845,845,845,845,845,816,466,234,879,954,93,655,167,875,875,2,878,958,351,399,93,231,919,919,919,919,88,305,184,184,932,184,564,564,340,877,51,717,305,717,717,717,638,251,324,324,521,521,521,858,875,875,875,840,840,840,840,840,840,840,752,752,752,752,628,628,628,628,946,946,946,946,946,946,946,946,213,329,675,675,958,958,958,958,958,13]

One person jump forward.

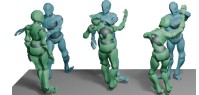

Pose codebobok Index

[144,544,544,953,108,562,723,443,532,238,464,383,726,813,60,60,238,238,383,383,238,413,503,503,503,299,299,870,637,531,45,110,663,234,234,369,184,584,805,161,837,635,571,507,398,927,927,631,894,542,927,688,981,0,556,992,936,166,315,295,674,877,400,811,1022,504,380,648,648,648,648,648,348,1019,796,546,563,327,414,198,198,737,737,198,637,134,132,620,769,698,769,769,556,720,601,601,906,396,101,315,315,490,56,210,324,271,271,877,43,582]

Two people shake hands.

| Person 1 Pose codebobok Index | Person 2 Pose codebobok Index |
|---|---|
| [31,486,424,496,16,106,723,544,953,953,93,144,663,776,986,986,953,953,953,986,986,953,686,168,960,266,120,232,888,464,688,688,688,968,93,953,233,1019,233,1019,233,233,953,613,323,203,243,203,888,215,89,528,528,941,952,941,941,941,89,89,941,941,941,941,941,941,259,124,968,3,940,450,450,782,923,782,782,450,607,299,450,782,299,782,407,407,407,407,382,31,912,734,734,538,538,668,286,917,917,299,269,286,286,286,299,538,450,450,32] | [52,1010,557,52,840,404,404,404,539,85,539,85,539,539,539,404,294,404,404,404,404,404,294,873,927,927,286,450,450,450,450,450,450,450,450,450,923,923,858,858,858,858,858,286,286,3,870,940,940,450,959,570,570,570,1019,1019,395,767,450,233,395,1019,1019,1019,1019,1019,161,746,746,746,674,786,592,786,786,786,656,516,516,516,450,516,786,786,786,786,786,786,786,223,223,465,465,466,223,516,450,485,485,516,516,538,538,516,895,538,538,538,538,450,615] |

Interaction codebobok Index

[196,685,76,908,474,980,980,492,673,492,492,221,196,673,673,673,492,673,673,673,673,492,566,324,234,773,381,21,257,257,257,257,257,558,558,810,773,291,488,291,488,488,810,275,1019,385,295,295,257,773,45,21,21,773,841,45,574,773,45,45,773,773,773,773,773,158,743,301,574,301,452,813,687,687,856,238,238,238,687,687,238,687,687,687,687,687,687,687,687,802,133,479,880,880,477,636,636,532,275,238,238,982,275,275,687,275,982,636,238,275,168]

Two people hug each other.

| Person 1 Pose codebobok Index | Person 2 Pose codebobok Index |
|---|---|
| [424,595,600,308,308,305,800,157,131,992,954,530,948,48,93,145,145,145,145,145,145,48,447,473,560,560,358,560,258,956,746,746,428,927,888,927,873,945,687,687,687,592,892,880,566,487,91,885,639,438,394,98,64,64,78,78,507,507,846,846,1019,1019,395,1019,1019,1019,119,181,111,271,947,668,938,938,734,144,792,144,168,792,281,281,281,59,62,59,59,498,601,526,281,682,935,134,536,433,934,934,934,934,934,934,934,934,934,934,934,390,934] | [157,131,761,177,400,964,896,29,155,28,893,281,646,458,458,458,771,28,771,118,411,893,562,891,271,485,88,395,250,316,463,233,449,7,395,91,7,31,233,562,7,562,7,233,251,251,245,86,858,450,858,878,98,9,919,450,769,401,944,83,485,83,83,83,166,335,888,395,885,653,885,885,799,919,919,675,675,263,263,263,13,13,13,263,263,675,263,13,554,223,213,223,258,481,462,560,530,193,762,193,193,193,193,193,193,193,193,193,193,193,193,193] |

Interaction codebobok Index

[522,431,110,422,496,534,298,903,633,262,797,620,669,985,985,985,35,985,35,871,596,596,22,651,857,601,535,774,563,105,542,335,695,95,841,853,299,134,677,946,142,677,142,693,517,254,157,291,22,595,713,133,654,24,905,692,362,362,713,184,291,291,291,101,464,464,329,329,445,544,544,532,555,555,69,201,69,201,830,69,1023,1023,69,69,69,69,1023,672,244,342,69,607,206,710,1019,130,1007,77,1007,1007,1007,1007,1007,1007,1007,1007,1007,1007,1007,1007]

Figure 10: Visual index of motions along with poses and interaction codes.

realism. This enables VR/AR systems and video games to provide more natural social interactions and immersive storytelling, enhancing user engagement and emotional presence.

- **Support for Assistive and Rehabilitation Technologies.** CODA can benefit elderly users or people with physical disabilities by generating realistic human actions for simulation, rehabilitation training, or intelligent prosthetic feedback. It can also help anticipate human needs and provide proactive support in smart environments.

