# OpenReview forum: "CODA: Commonsense-Driven Autoregressive Human Interaction Generation"
_ICLR.cc/2026/Conference — ICLR 2026 Conference Withdrawn Submission_

### Official Review · Reviewer_rWNy · 2025-10-27

**Soundness:** 1
**Presentation:** 3
**Contribution:** 2
**Rating:** 2
**Confidence:** 4

**Summary:**

This paper aims to design a plug-and-play model to address the ``commonsense'' issues in two-person interactions. Specifically, it imposes constraints on individual joint trajectories, adjusts the center of mass, and enforces distance and collision constraints during multi-person interactions. These measures effectively suppress motion artifacts and explicitly enhance commonsense plausibility. Furthermore, the model employs VAE and Codebook techniques to circumvent the need for a simulator in solving these problems.

**Strengths:**

**Originality:** This paper proposes a novel approach that avoids the use of a simulator to address the commonsense issues in kinematics-based motion generation. The idea of achieving physically plausible motion through purely kinematic methods is highly original.

**Quality:** The paper is well-written and accessible. The figures and tables are clear and effectively support the content.

**Clarity:** The description of the proposed method is clear and easy to follow.

**Significance:** Improving the quality of two-person interaction motions and eliminating physical artifacts is an important and urgent problem in the field.

**Weaknesses:**

**1. Lack of Definition and Evaluation of ”Commonsense''**

The paper claims to address the ``commonsense`` issue in both single-person motion generation and two-person motion interaction. However, **the term ``commonsense`` is neither clearly defined nor rigorously discussed throughout the manuscript. Furthermore, no novel evaluation metric is proposed to assess the quality or validity of the so-called commonsense in the generated motions.** This omission significantly undermines the scientific rigor of the claimed contribution.

**2. Novelty and Adequacy of the Commonsense Constraint Loss (CCL)**

The proposed Commonsense Constraint Loss (CCL) appears to lack sufficient novelty to be considered a core contribution. The authors attempt to address several physical implausibilities, including bone stretching, penetration, single-person joint incorrectness, center-of-mass (CoM) position incorrectness, and two-person interaction distance incorrectness. However, **similar regularization strategies have already been introduced in InterGen (Section 4.3, ``Additional Regularization Losses``)**, where constraints on geometric and interactive aspects are explicitly enforced.

Moreover, **the adjustment of the center-of-mass position seems overly simplistic**. If the goal is to achieve commonsense-level physical plausibility, shouldn't different body parts have their own respective centers of mass? In the supplementary video ``Supplementary materials/demo_videos/Compare_results/our_hug.mp4``, a noticeable positional error in the head is observed. This raises the question: Is the current granularity of CoM adjustment too coarse to meaningfully capture commonsense physical correctness?

**Summary**

Based on the lack of a clear definition and explanatory depth around the claimed core contribution of "commonsense", as well as the insufficient improvements made in this regard and the limited novelty of the proposed CCL loss, I recommend that the authors further investigate these issues and provide comprehensive and effective solutions. If the motion quality during the kinematics-based generation stage is still unsatisfactory, is it really necessary to "avoid using a simulator" at this early point?

**Questions:**

**Suggestions**

1. As mentioned in ``Weaknesses``, please clearly give the definition of ”commonsense'', and re-evaluate the significance of this task.
2. Provide additional metrics to clarify the ”commonsense'' gap between the previous work and this paper.

**Questions**

1. The $L_{KTraj}$ and the $L_{COM}$ both effects pelvis joint; could they conflict? I’m a bit confused. Please briefly explain it.
2. The paper encodes two-person interaction features into a high-level codebook—does this scale? The InterHuman dataset still focuses on a set of specific scenes; will the ICS method remain robust as it grows? With virtually infinite ways for two people to interact, maintaining a codebook seems unlikely to cover ever-expanding scenarios. If an ablation study could be provided to demonstrate that ICS remains effective as scenarios continue to expand, it would further enhance the credibility of the ICS approach.

---

### Official Review · Reviewer_n1gh · 2025-10-28

**Soundness:** 4
**Presentation:** 4
**Contribution:** 3
**Rating:** 6
**Confidence:** 2

**Summary:**

This paper proposes CODA, a commonsense-driven autoregressive framework for text-to-interaction human motion generation.
The core idea is to explicitly model commonsense plausibility — ensuring that generated interactions obey physical and geometric intuition (e.g., avoiding penetration, maintaining balance). CODA introduces two key novel components:
(1) an Interactive Codebook Storager (ICS) that separately encodes pose-level and interaction-level representations via two codebooks (C and P); and
(2) a Commonsense Constraint Loss (CCL) composed of four terms — center-of-mass, key-joint trajectory, Gaussian distance map, and penetration penalty — each designed to enforce human-plausible motion.
A Conditional Masked Transformer then generates interactions autoregressively in latent space.
Extensive experiments on InterHuman show that CODA improves FID, R-Precision, and physical plausibility over diffusion- and autoregressive-based baselines.

Overall, this work makes a solid and practically relevant contribution: the methodology is clear, well-motivated, and effectively validated. While the novelty is moderate, the approach is conceptually valuable for advancing commonsense-aware human interaction generation.

**Strengths:**

1. Meaningful framework and methodology.
The introduction of separate pose and interaction codebooks, together with CCL, is intuitive and well-justified. The two-stage architecture (CVQ-VAE + Conditional Masked Transformer) is clearly described, with explicit mathematical formulations. The CCL is modular and can be easily integrated into other generative frameworks.

2. Comprehensive and convincing experiments.
Quantitative and qualitative evaluations on InterHuman demonstrate consistent improvements in FID, R-Precision, and plausibility metrics (PBC↑ / IP↓). Ablation studies (Tables 4–7) are detailed, effectively isolating the contributions of single-person and interaction-level constraints.

3. Reproducibility and transparency.
The appendix provides extensive implementation details, ablation protocols, and reproducibility notes. The planned code release further strengthens reliability.

**Weaknesses:**

1. The paper highlights the importance of commonsense plausibility, and this is a valuable direction. Still, the surrounding discussion could be expanded to better situate the contribution within existing perspectives. Commonsense in motion generation is ultimately evaluated through metrics, but the paper does not yet explore in depth what “commonsense” represents or how it might be systematically measured. There are also several works on single-human motion that could serve as useful references for understanding or quantifying this aspect [1] [2] [3]. These studies view commonsense primarily through perceptual or physics-informed lenses, which might provide complementary insights to the loss-based perspective adopted here. Offering a brief discussion of how CCL relates to or differs from such approaches could make the contribution more contextualized and connected to prior work.

[1] What is the Best Automated Metric for Text-to-Motion Generation? (SIGGRAPH 2023)

[2] Aligning Human Motion Generation with Human Perceptions (ICLR 2025)

[3] PP-Motion: Physical–Perceptual Fidelity Evaluation for Human Motion Generation (ACM MM 2025)

2. At the same time, while the CCL and its four components (LCOM, LKTraj, LGDM, LPPL) are thoughtfully designed and empirically validated, they may not yet capture the full scope of “commonsense plausibility.” The current formulation mainly reflects inductive biases about geometry and physics, whereas commonsense in human interactions can also involve perceptual and social expectations [4] [5]. The work might be further strengthened by briefly discussing how such perceptual priors—or even off-the-shelf perceptual models—could complement the existing inductive-bias-based losses, leading toward a more comprehensive notion of commonsense guidance.

[4] Social Motion Prediction with Cognitive Hierarchies (NeurIPS 2023)

[5] in2IN: Leveraging Individual Information to Generate Human Interactions (HuMoGen CVPRW 2024)

3. The connection between ICS and CCL seems loose. It seems that these are 2 separate techniques, and while the framework claims that CCL guides ICS to learn commonsense-plausible features, the empirical link is not fully demonstrated. It is unclear whether ICS itself inherently improves commonsense plausibility, or it's the loss that works, beyond providing better latent representations as visualized in Fig. 5.

4. A few presentation and reproducibility details could be refined for clarity and consistency.

 • In Tables 1 and 6, the diversity values appear to be bolded inconsistently; Table 5 (row B) also lists a potentially incorrect diversity value (0.767).

 • Some visualization examples show unnatural head orientations; adpoting InterMask style visualization style might improve clarity and comparability.

**Questions:**

1. On evaluating commonsense plausibility

Commonsense plausibility involves not only adherence to physical laws but also alignment with natural human interaction patterns — a very challenging aspect to quantify. How do the authors think about defining or evaluating this concept more systematically? In particular, what would be a reasonable metric or evaluation protocol to measure “commonsense” beyond current indicators such as FID, PBC, etc.

2. On extending to multi-human scenarios

Would it be feasible to extend the current framework to multi-human interactions (e.g., three or more participants)? What potential bottlenecks might arise — for instance, dataset limitations, pairwise modeling assumptions, or the design of the commonsense constraints? It would also be interesting to know whether the authors have attempted zero-shot generation in such settings. From another perspective, if extending to multi-human interactions would require rethinking some components of CCL, it might suggest that the current formulation, while strong, represents an intermediate step rather than advancing towards final solution.

3. On loss weighting and sensitivity

How were the loss weights (λCOM, λKTraj, λGDM, λPPL) determined in practice? Were they chosen empirically, via validation tuning, or based on heuristic balancing? It would be helpful to understand whether CODA’s performance is sensitive to these hyperparameters or relatively stable under moderate changes.

I am open to hearing the authors' opinions and willng to raise my rating if my concerns are addressed.

---

### Official Review · Reviewer_ZWaJ · 2025-10-29

**Soundness:** 2
**Presentation:** 2
**Contribution:** 2
**Rating:** 2
**Confidence:** 4

**Summary:**

In this paper, the authors proposes CODA, a two-stage autoregressive framework for generating commonsense-plausible human interaction motions from textual descriptions. Specifically, the CODA approach introduces two key components: the Interactive Codebook Storager (ICS), which stores motion features for both single-person and two-person interactions, and the Commonsense Constraint Loss (CCL), which enforces physical and geometric plausibility through hierarchical constraints. The model is trained on the InterHuman and KIT-ML datasets and evaluated across multiple metrics, showing some improvements over state-of-the-art baselines in realism, semantic consistency, diversity, and physical plausibility.

**Strengths:**

1. CODA introduces a unique combination of vector-quantized autoencoding with commonsense-guided constraints, offering a plug-and-play solution.
2. A dual-codebook system (ICS) for storing individual and interaction motion features.
3. A commonsense constraint loss that enforces physical and geometric plausibility via hierarchical constraints and integrates multiple constraints to suppress motion artifacts.
4. Evaluated on InterHuman and KIT-ML datasets using metrics like FID, R-Precision, MPJPE, and Diversity.

**Weaknesses:**

1. CODA introduces a dual-codebook architecture (pose and interaction) and a commonsense-guided constraint loss (CCL) composed of four handcrafted components. While conceptually well-motivated, this rigid formulation may limit adaptability to diverse motion styles or unseen interaction types. The reliance on fixed hyperparameters and manually designed loss functions (e.g., COM, KTraj, GDM, PPL) constrains the model’s flexibility and generalization to varied social dynamics.
2. The Gaussian Joint Distance Map (GDM) loss employs a soft-thresholding mechanism based on a fixed Gaussian kernel (τ = 0.5) to emphasize close joint pairs. However, this heuristic design lacks adaptivity and may be sensitive to the choice of τ. The absence of a learnable or data-driven thresholding strategy raises concerns about robustness across datasets with different spatial scales or interaction densities.
3. The Penetration Penalty Loss (PPL) relies on Axis-Aligned Bounding Boxes (AABB) and minimum bone-to-bone distances to detect interpenetration. This method may struggle in complex or occluded poses where AABB approximations are insufficient. Moreover, the use of a fixed 0.9 scaling factor in the penalty term is not empirically justified, leaving its generalizability across body types and motion scales questionable.
4. The paper introduces a dual-codebook system as a core architectural component but does not provide interpretability analysis or empirical insights into how these codebooks are populated or utilized during inference. Without understanding codebook diversity, redundancy, or semantic structure, it is difficult to assess whether the system learns meaningful motion primitives or merely memorizes patterns.
5. The fusion of individual and interaction features via simple vector summation (wa = za + ta) is a minimalistic design choice. The paper does not explore any alternatives such as attention-based fusion or gating mechanisms.
6. The Conditional Masked Transformer (CMT) uses an iterative refinement strategy involving remasking and resampling of low-confidence tokens. While this improves motion quality, it introduces significant computational overhead during inference. The paper does not provide any analysis of inference time, memory usage, or scalability, making it difficult to compare.
7. Although the model uses autoregressive generation and masked token prediction, it lacks explicit mechanisms to enforce temporal smoothness or continuity. This could lead to jittery or discontinuous motion, particularly in long sequences or transitions between interaction phases, which are common in real-world applications.
8. All comparisons on the InterX dataset are missing. Given that InterX is a widely used benchmark for evaluating interactive motion generation, the absence of results on this dataset limits the ability to contextualize CODA’s performance relative to prior work.

**Questions:**

1. InterX is a widely adopted benchmark for evaluating interactive motion generation. Could the authors clarify why results on InterX were not included? Including these results may strengthen the paper by better contextualizing CODA’s performance relative to prior works.
2. Could the authors elaborate on how the fixed formulations and hyperparameters in the four handcrafted components of CCL (COM, KTraj, GDM, PPL) affect the model’s adaptability to diverse motion styles or unseen interaction types? Have they considered or tested any learnable or adaptive alternatives?
3. Both GDM and PPL rely on fixed heuristic parameters (e.g., τ = 0.5 in GDM, 0.9 scaling in PPL). Could the authors justify these choices and discuss their sensitivity across datasets? Will the adaptive or learnable alternatives be better to improve robustness, especially in complex or occluded interactions?
4. Can the authors provide insights into how the pose and interaction codebooks are populated and utilized during inference?
5. What motivated the choice of simple vector summation (wa = za + ta) for fusing individual and interaction features? Have the authors explored or compared more expressive alternatives such as attention-based fusion or gating mechanisms?
6. Could the authors provide an analysis of the computational cost (e.g., inference time, memory usage)?
7. How does CODA ensure temporal smoothness or continuity across frames, especially in long sequences or transitions between interaction phases? Have the authors observed any jitter or discontinuity in generated motions, and if so, how is it mitigated?

---

### Official Review · Reviewer_99Fk · 2025-10-31

**Soundness:** 3
**Presentation:** 3
**Contribution:** 2
**Rating:** 4
**Confidence:** 4

**Summary:**

This study addresses the task of human interaction generation with an emphasis on commonsense plausibility. Building upon a baseline masked autoregressive model, the authors introduce two codebooks designed for single-person and two-person motion generation. To promote more realistic and physically consistent interactions, several regularization losses, center of mass loss, key joint trajectory loss, Gaussian joint distance map loss, and penetration penalty loss, are incorporated into the VQ-VAE training process. Experimental results demonstrate that the proposed method achieves improved performance over the baseline model.

**Strengths:**

1. The paper is well written and clearly structured, making it easy to follow. The figures effectively illustrate the main concepts and aid comprehension.
2. The authors conduct extensive experiments, providing both qualitative and quantitative evaluations.
3. The proposed loss functions appear to meaningfully mitigate issues of unnatural poses and contact artifacts in human interaction motion generation.

**Weaknesses:**

1. Compared with prior work in human interaction generation, this approach primarily introduces additional regularization losses to the motion generation framework. While effective, the contribution appears to be an incremental engineering enhancement rather than a substantial methodological or theoretical innovation.
2. The paper claims that the commonsense plausibility compound loss is plug-and-play. However, as described, the loss functions must be integrated during model training rather than applied independently during inference. While they are compatible with other baseline models, the term “plug-and-play” typically refers to modules that can be directly used without retraining. Therefore, this phrasing is somewhat misleading.
3. The qualitative results demonstrate improvements in generated interactions, but these examples could be selectively chosen. For quantitative evaluation offering statistical assessment, however, most metrics, except for FID, show only marginal improvement over the InterMask baseline. It would strengthen the work to include statistical significance testing to confirm the observed gains.
4. The paper suggests that the key joint trajectory loss improves motion realism, yet the supplementary videos show limited improvement in foot motion consistency. A more detailed analysis of this component’s effectiveness would be helpful.
5. The evaluation is conducted on a single dataset. To further validate generalizability and robustness, it is recommended to include experiments on a larger dataset such as Inter-X to better demonstrate the superiority of the proposed approach.

**Questions:**

1. For quantitative evaluation offering statistical assessment, however, most metrics, except for FID, show only marginal improvement over the InterMask baseline. Is the improvement statistical significance?
2. The paper suggests that the key joint trajectory loss improves motion realism, yet the supplementary videos show limited improvement in foot motion consistency. Is there any reason for this observation?

---

### Note · Authors · 2025-11-13

I have read and agree with the venue's withdrawal policy on behalf of myself and my co-authors.